# Antioxidant Activity and Kinetic Characterization of *Chlorella vulgaris* Growth under Flask-Level Photoheterotrophic Growth Conditions

**DOI:** 10.3390/molecules27196346

**Published:** 2022-09-26

**Authors:** Jesús Alberto Coronado-Reyes, Evelyn Acosta-Ramírez, Miranda Valeria Martínez-Olguín, Juan Carlos González-Hernández

**Affiliations:** Tecnológico Nacional de México/Instituto Tecnológico de Morelia, Av. Tecnológico 1500, Col. Lomas de Santiaguito, Morelia 58120, Michoacán, Mexico

**Keywords:** photoheterotrophic, flask, sodium acetate, glycerol, photoperiod, antioxidant activity

## Abstract

*C. vulgaris* is a unicellular microalgae, whose growth depends on the conditions in which it is found, synthesizing primary and secondary metabolites in different proportions. Therefore, we analyzed and established conditions in which it was possible to increase the yields of metabolites obtained at the flask level, which could then be scaled to the photobioreactor level. As a methodology, a screening design was applied, which evaluated three factors: type of substrate (sodium acetate or glycerol); substrate concentration; and exposure-time to red light (photoperiod: 16:8 and 8:16 light/darkness). The response variables were: cell division; biomass; substrate consumption; and antioxidant activity in intracellular metabolites (ABTS•+ and DPPH•). As a result, the sodium acetate condition of 0.001 g/L, in a photoperiod of 16 h of light, presented a doubling time (Td = 4.84 h) and a higher rate of division (σ = 0.20 h^−1^), having a final biomass concentration of 2.075 g/L. In addition, a higher concentration of metabolites with antioxidant activity was found in the sodium acetate (0.629 Trolox equivalents mg/L ABTS•+ and 0.630 Trolox equivalents mg/L DPPH•). For the glycerol, after the same photoperiod (16 h of light and 8 h of darkness), the doubling time (Td) was 4.63 h, with a maximum division rate of σ = 0.18 h^−1^ and with a biomass concentration at the end of the kinetics of 1.4 g/L. Sodium acetate under long photoperiods, therefore, is ideal for the growth of *C. vulgaris*, which can then be scaled to the photobioreactor level.

## 1. Introduction

The study of microalgae has gained importance in recent years, because microalgae are considered as raw materials for chemical compounds that result from their primary and secondary treatment [1]. Examples of primary metabolites include: lipids to generate biodiesel [2,3,4,5]; proteins for the formulation of food supplements [6,7] or functional foods [8,9]; and carbohydrates that are used as a source of fiber for food [10,11]. On the other hand, there are secondary metabolites [12,13,14], among which, in *C. vulgaris*, are pigments such as chlorophyll [15], which constitutes 1 to 2% of the dry weight of the biomass. In addition to these pigments, there is also the presence of carotenoids, that belong to the group of secondary metabolites called terpenes. The carotenoids identified in microalgae are essentially β-carotenes that associate with lipids present in chloroplasts, as well as with chlorophylls and thylakoids in the chloroplasts themselves. The identified pigments have been evaluated for their antioxidant activity [16] and have been reported to have been extracted using solvents such as dimethylformamide, dichloromethane, acetone, hexane and ethanol, by methods such as Soxhlet, ultrasonically assisted extraction and supercritical fluids. Among the main pigments that have been identified in *C. vulgaris* are β-carotenes, astaxanthin, canthaxanthin, lutein, chlorophyll a and b, pheophytin a and b and violoxanthin. As can be seen, *C. vulgaris* contains polyphenols such as luteolin, so it would be possible to use other solvents, with which it can be evaluated and characterized, if there are others [3]. Regarding polyphenols in *C. vulgaris* [17], it has recently been reported that there is a presence, and that it is possible to have a concentration of 217 mg/100 g of dry sample in aqueous extracts, and a concentration of 109 mg/100 g of dry sample in ethanolic extracts. The above was established when performing the maceration of lyophilized biomass with polar solvents such as water and 96% *w*/*w* ethanol. Polyphenols were quantified using the *Folin-Ciocalteu* method, using gallic acid as a positive control stock, and expressing the results in gallic acid equivalents. In addition to this, the antioxidant activity was evaluated by the 2,2′-azino-bis-3-ethylbenzothiazoline-6-sulfonic acid (ABTS•+)-and-2,2-diphenyl-1-picrylhydrazyl (DPPH•) method. For ABTS•+ there was an inhibition only in the aqueous extracts of 83–93%, and for DPPH• there was an inhibition in the same extracts of 80.1%, so it was interesting to expand our knowledge about the extraction of these metabolites, and how to increase the yields that currently exist, as they are generally used in the food industry or in medicine, and extracted from other materials such as fruits, stems, leaves and flowers [14]. Previous to this study, one can also cite the study entitled ‘Antioxidant Potential of Extracts Obtained from Macro (*Ascophyllum nodosum*, *Fucus vesiculosus* and *Bifurcaria bifurcata*) and Micro-Algae (*Chlorella vulgaris* and *Spirulina platensis*) Assisted by Ultrasound’, in which 5 g of biomass of each species was used, to be subjected to ultrasound extraction in a 50:50, *v*/*v*, water/ethanol solution, in a final volume of 50 mL. The antioxidant activity was evaluated with the ABTS•+, DPPH•, Oxygen Radical Absorbance Capacity Assay (ORACA) and Ferric Reducing Antioxidant Power (FRAP) methods, and the highest response was found in macroalgae, specifically in *Bifurcaria bifurcata*, being 537.74, 146.27, 575.02 and 67.4 µmol Trolox equivalents/g dry sample, respectively. For *C. vulgaris*, the highest response was found in microalgae, being 15.64, 0.86, 33.07 and 0.62 mol equivalents of Trolox/g dry sample, respectively; therefore, the president believed that these microalgae were ideal to continue with his research [18]. The main reason for using microalgae at an industrial level is because their cultivation is easy and fast, as they complete their growth cycle in a few days, and can grow in salt water, fresh water, wastewater and sometimes in fermentation tanks [19]. *C. vulgaris* is a kind of microalgae that can grow easily and quickly. Of all the known species of microalgae, *C. vulgaris* is one of the most easily adapted, and its metabolism can be photoautotrophic, photoheterotrophic or mixotrophic in open or closed culture systems [20,21,22]. It has been observed that the highest biomass concentration of *C. vulgaris* is generated when working under mixotrophic conditions at flask and photobioreactor level [1,21]. Therefore, it is interesting to investigate the growth of *C. vulgaris* under photoheterotrophic conditions at the flask level, to analyze the factors that have a significant effect on the growth of *C. vulgaris*, and to subsequently perform the growth at the photobioreactor level. Within the growth conditions that have been investigated with respect to the type of substrate, the one entitled ‘Effect of Glycerol and Glucose on the Enhancement of Biomass, Lipid and Soluble Carbohydrate Production by *Chlorella vulgaris* in Mixotrophic Culture’ is notable, where it is observed that *C. vulgaris* can grow with glycerol as a substrate, but to a lesser extent when using a mixture of glycerol and glucose. It is important, however, to find conditions in which glucose can be replaced, as this substrate makes it an expensive process. For this study, the growth was carried out under concentrations of 1, 5 and 10 g/L of glycerol, to which 2 g/L of glucose were added, under a photoperiod of 12 h of light and 12 h of darkness, in a volume of 100 mL of medium for 96 h (4 days) [23]. Substrate concentrations and even type of photoperiod were varied, to increase the growth and synthesis of metabolites. In regard to the photoperiod for the photoheterotrophic cultivation of microalgae, the light source may be provided by the sun or an artificial light source, but only certain wavelengths of light, ranging from 400 to 700 nm are effective for productive algal photosynthesis. In addition, different species respond differently to light wavelengths, due to variations in their pigments. It has been observed, however, that red light, with a narrow spectrum of 600–700 nm, is the optimal wavelength for the growth of most algal species. This is primarily because the most abundant pigments in most species are chlorophylls that can more efficiently absorb red light compared to other light wavelengths. Light with a shorter wavelength—for example, blue light—has a higher probability of causing photo-inhibition by striking the light-harvesting complex of cells at its peak of electrical energy, due to its high energy. Due to its longer wavelength, however, the lower light energy of red light inhibits its ability to penetrate high-density or deep cultures. For this reason, the use of red light is recommended. Recently, a study entitled ‘Cultivation of *Chlorella vulgaris* in Wastewater with Waste glycerol: Strategies for Improve Nutrition Removal and Enhance lipid Production’ was published, in which the same concentrations of pure glycerol and pre-treated glycerol were used; growth was observed at 7 days, when it was shown that, for both conditions, the best growth was at a concentration of 5 and 10 g/L of pure and pre-treated glycerol, with a significant accumulation of lipids in the microalgae [24]. Regarding growth with sodium acetate, the study entitled ‘Effect of various carbon sources on the production of biomass and lipids of *Chlorella vulgaris* during conditions of nutrient scarcity and nitrogen scarcity’ is notable, in which the productivity of biomass of 0.158 ± 0.011 g/L/d was achieved using sodium bicarbonate, followed by 0.130 ± 0.013, 0.111 ± 0.005 and 0.098 ± 0.003 g/L/d for sodium acetate, carbon dioxide and molasses, respectively. The use of carbon dioxide resulted in the highest cell density, while the use of sodium acetate resulted in the highest content of fatty acids. *C. vulgaris* microalgae therefore have generated a large field of research and many applications regarding the metabolites that they synthesize, and the type of growth condition for their study. Little research has been done, however, on secondary metabolites, and how these can increase their synthesis depending on growth conditions, as previous studies have focused on the products of primary metabolism [25,26]. Thus, investigating the presence and optimal production of these molecules with this activity can contribute to the development of alternatives for the energy, pharmaceutical and food sectors, which was the main objective of the following research.

## 2. Results

As the first results to have been analyzed, cell division is presented in the following Figure 1, to show what kind of substrate and what concentration and photoperiod favored the generation of new cells, starting from an initial concentration of 5.0^7^ cells/mL.

Based on Figure 1, it can be clearly observed that under the sodium acetate condition, and under the three concentration conditions, there was an increase in the number of initial cells reaching the exponential phase around hour 96, entering the stationary phase around hour 144, and taking an adaptation time of around 72 h, on which basis, it can be said that this growth condition allowed *C. vulgaris* to activate its metabolism after a period of 3 days. Comparing the photoperiod with sodium acetate under 8 h of light, with respect to the three conditions of substrate concentration, the highest value in Log_10_ was obtained at hour 168, being 2.16, 2.02 and 1.96 for the concentrations of 0.0005, 0.001 and 0.0015 g/L, respectively; after performing a one-way ANOVA test, it was found that, at this time, there were significant differences, and that the best condition was under a concentration of 0.0005 g/L. The value of r^2^ was 0.998. For the photoperiod of 16 h of light at the same time, 168 was where the highest value of Log_10_ was found, being from 2.37, 2.37 and 2.52 for the concentration of 0.0005, 0.001 and 0.0015 g/L, respectively. The first two conditions, however, were statistically the same, with only the highest concentration being different. The r^2^ value for this one-way ANOVA was 0.999. It can therefore be said that, for this photoperiod, there was an equally positive effect for the lowest and medium conditions in the substrate concentration, but that for a higher concentration of substrate, there was inhibition of cell growth.

Finally, to analyze if the type of photoperiod influenced cell growth, the points with the maximum cell growth at the same time of growth (168 h) were taken into account, and it was observed that, statistically, the conditions were different, except for the condition of 16 h of light and 8 h of darkness at a substrate concentration of 0.0005 and 0.001 g/L of sodium acetate; however, these were the conditions with the highest cell growth, and so it can be said that the type of photoperiod influenced cell growth and that those which had a greater time prolongation favored cell growth in *C. vulgaris.* The *Prob > F* value was <0.0001, with an ANOVA r^2^ value of 0.999. In regard to the glycerol condition, only under the condition of 0.27 g/L after a photoperiod of 8 h of light and 16 h of darkness was there significant growth, beginning the exponential passage at hour 96, and reaching the stationary phase at hour 168. In regard to the photoperiod of 16 h of light and 8 h of darkness, there was growth both in the 0.27 and 0.33 g/L conditions, beginning the exponential phase at hour 72, and reaching the maximum cell growth at hour 168, when the stationary phase was observed, so that it can be said that saturation in the medium with glycerol at 0.44 g/L inhibited cell growth. In regard to the comparative analysis, through the one-way ANOVA and *Tukey-Kramer HSD* analysis, the value of r^2^ for the ANOVA was 0.994 with a *Prob > F* value of <0.0001, so there were significant differences. With the *Tukey-Kramer HDS* analysis, it was possible to identify that the glycerol conditions at a concentration of 0.33 and 0.44 g/L after a SHORT photoperiod (8 h of light), and the condition of 0.44 g/L after a PROLONGED photoperiod (16 h of light), were not significantly different, as there was no cell division. The condition that favored growth, however, was that of 0.33 g/L of glycerol in prolonged photoperiods. Furthermore, the rest of the experimental conditions were statistically different; therefore, the type of photoperiod, as well as the concentration and kind of substrate, are relevant for the growth of *C. vulgaris*.

In this study, we were looking for an increase in the number of new cells formed, as this was a study intended to serve as a background in growth kinetics at the photobioreactor level; in this way, with a greater number of cells, it will be possible to have a greater concentration of synthesized metabolites, if the factors can be found that promote said synthesis, as there may be large cells (biomass), but they do not necessarily have a greater synthesis of pigments. The foregoing is evident from what was observed in a study by Kong et al. from the year 2020, in which there was an increase in the amount of biomass generated, but significant decreases in the number of pigments in *C. vulgaris* [23].

For the kinetics, it was decided to start with a concentration of 50^7^ cells/mL, as the kinetics were started with concentrations of 10^7^, 20^7^ and 30^7^ cells/mL in previous exploratory studies but, at the time of sampling, no significant amount was observed. There was a considerable number of cells in the Neubauer chamber, so the explore sample takings indicated a lower concentration than the initial ones, and there was too much variation in the cell count. With a concentration of 50^7^ cells/mL, this count was better observed. Figure 1B,D show the behavior in the kinetic phases, as the effect of the photoperiod was observed. We believe, however, that the number of cells was not greater because the inorganic substrate was limiting, and this was reaffirmed as shown in Figure 2. For this study, optimization conditions were sought, i.e., a low amount of substrate, but enough to promote cell growth, as well as to promote the synthesis of metabolites due to stress condition.

On the other hand, the biomass formed. Figure 2, below, shows that the kinetic trace was not completely carried out.

Figure 2A,B show that a decrease in biomass was observed with respect to the initial biomass, which may have been due to the fact that, if compared to Figure 1, cell growth occurred, but not a growth or enlargement of the cells individually; therefore, under the dry weight technique, high sensitivity was not allowed where the small and numerous new cells formed were detected for their weight. In addition to this, it was observed that during the days of kinetics, the medium turned green, and there were sections where a grouping of cells in the form of clusters occurred, so that at the time of taking the sample for dry weight, there was no uniform sampling of cellular material, even though growth occurred under agitation. With the data obtained, however, it was possible to calculate the amount of biomass in the flask, and the degree to which it could be concentrated in order to formulate and produce biotechnological products, as the formation of cells can be seen in Figure 1.

Taking into account the biomass after the exponential phase of Figure 1, it was found that the highest concentration was reached in sodium acetate after 8 h of light, at hour 144, close to the hour where the stationary phase began, and where there was a maximum concentration of 0.85, 1.15 and 1.03 g/L for the concentration of 0.0005, 0.001 and 0.0015 g/L of substrate, respectively. For sodium acetate at 16 h of light, the maximum concentration was reached at hour 168, being 1.63, 2.03 and 2.4 g/L for the concentration of 0.0005, 0.001 and 0.0015 g/L of substrate, respectively. With these data, a one-way ANOVA analysis was performed, which had an r^2^ of 0.991, indicating significant differences between the conditions evaluated; with the *Tukey-Kramer HSD* test, significant differences were observed, in which sodium acetate at a photoperiod of 16 h of light achieved the maximum concentration, and with which it can be seen that the type of photoperiod and the type of substrate had a greater effect than the concentration of the substrate. Despite this, a maximum concentration of biomass was found at hour 24, after a photoperiod of 16 h of light, which was 3.825 g/L, so it would be interesting to see if new parameters could be varied at this time that would allow for increase of this value or for maintaining it throughout the experimental kinetics.

Regarding the behavior of the biomass for glycerol in Figure 2C,D, it can be seen that there was no ascending line in the amount of biomass, both for the photoperiod of 16 h of light, and for the photoperiod of 8 h of light, as in both cases the values ranged between a concentration of 1.2 g/L to 2.3 g/L; a *Tukey-Kramer HSD* test indicated that there were no significant differences, so that in glycerol, the type of substrate concentration had no effect, nor the type of photoperiod.

The consumption of substrate within growth kinetics behaved inversely to the generation of new cells, biomass and products; Figure 3 shows the observed behavior:

As shown in Figure 3A,B, there was a significant consumption in the different concentrations of substrate. In Figure 3B, a greater decrease can be seen. This behavior was in agreement with that observed in Figure 1B, which was the experimental condition with the greatest formation of cells. In Figure 3A, the decrease was less pronounced, as well as the minor increases in the cells of Figure 1A. Based on these observations, we concluded that the substrate used was being consumed by the microalgae, to activate its metabolism and promote its cell division; when the sodium acetate began to run out or be limited was when the microalgae entered its stationary state. About the explanation as to the behavior being constant or slow at hour 144 was in Figure 1A, due to whether there was already a decrease in the consumption of substrate, it can show in Figure 3A; the behavior coincided with the disposition of the diluted substrate: when less was available for the microalgae, its exponential growth stopped going up, and there was not even a marked exponential phase, as observed in Figure 1B. In the same way, this applied to the effect of the photoperiod, as it was an inducing factor in metabolic activity for cell growth. In the case of the 0.0005 g/L concentration, at hour 144 the substrate availability was minimal, how will it show in Figure 3A; this low value may have been due to the lack of consumption of sodium acetate, and thereby growth was not promoted.

With regard to glycerol consumption, it can be seen from Figure 3C,D, that it was only in the condition of a prolonged photoperiod (Figure 3D) that a decrease in glycerol concentration occurred, and that this same behavior was reversed, in regard to the number of cells formed, in Figure 1D. Regarding Figure 3C (short photoperiod), it can be seen that in the glycerol conditions of 0.001 and 0.0015 g/L, the decreases were minimal, as there were no conditions that favored inducing metabolic activity in the microalgae, as shown in Figure 1C, and starting the consumption of substrate, so that the glycerol remained constant or had slight variations in concentration throughout the kinetics. With the above, the fact that the type of photoperiod, as well as the type of substrate, directly influenced metabolic activation for cell growth was reaffirmed.

Regarding the products formed with antioxidant activity, Figure 4 and Figure 5 are presented below. Figure 4 shows the antioxidant activity evaluated by the ABTS•+ method, and Figure 5 shows the results corresponding to the activity evaluated under the DPPH• method.

Figure 4 shows a behavioral trend in the data similar to that shown in Figure 1; therefore, it can be assumed that, as the number of cells increased, there was an increase in the amount of metabolites with antioxidant activity synthesized and stored. A bivariate analysis was carried out, through a *Pearson’s correlation*, which established that sodium acetate, after a photoperiod of 8 h of light and 16 h of darkness, had an r^2^ value of 0.46, 0.37 and 0.17 for sodium acetate concentrations of 0.0005, 0.001 and 0.0015 g/L, respectively; a correlation could therefore be seen, but it was not linear, which may be due to the fact that the synthesis of other compounds—products of the primary metabolism—was favored, and that this experimental condition did not cause the microalgae to synthesize a great concentration of secondary metabolites with this activity, which protected the cell when it was under stress. Regarding what was observed in sodium acetate after a photoperiod of 16 h of light and 8 h of darkness, an r^2^ value of 0.91, 0.85 and 0.05 was obtained for the concentration 0.0005, 0.001 and 0.0015 g/L, respectively, which indicated that this stress situation promoted an increase in the amount of secondary metabolites that protected the cell from light exposure. The condition where a better correlation was seen was under a concentration of 0.0005 g/L, as well as 0.001 g/L, of acetate; therefore, it can be said that these were ideal for inducing the synthesis of metabolites with antioxidant activity.

Regarding glycerol as a substrate, *Pearson’s correlation* showed that after a photoperiod of 8 h of light, the value of r^2^ of the ratio of secondary metabolites, with respect to the number of cells formed, was 0.75, 0.05 and 0.02 for the concentration of 0.27, 0.33 and 0.44 g/L, respectively; therefore, only the 0.27 g/L condition showed a positive response in the formation of these metabolites. The rest of the experimental runs yielded few data, as there was no cell growth under those conditions. On the other hand, after a photoperiod of 16 h of light and 8 h of darkness, the following values of r^2^ were obtained: 0.80, 0.15 and 0.07 for the concentration of 0.27, 0.33 and 0.44 g/L, respectively; therefore, something similar to what was observed in the short photoperiod occurred; however, it can be seen that there was greater linearity in the 0.27 g/L condition of glycerol, which was therefore ideal for favoring the formation of these products when glycerol was used as a substrate.

Finally, regarding Figure 5, the correlations between the antioxidant activity and the number of cells formed in the different experimental conditions were similar to those indicated in the previous paragraphs; however, the results obtained between the antioxidant activity measured through the ABTS•+ and DPPH• method differed, because the first measured lipophilic and hydrophilic metabolites, while the second measured only lipophilic metabolites. For the sodium acetate condition, after a short photoperiod (8 light hours), the value of r^2^ was 0.82, 0.59 and 0.55 for the concentration of 0.0005, 0.001 and 0.0015, respectively; therefore, again, the condition that allowed for better linearity under stress was the condition of 0.0005 g/L of sodium acetate. As for the long photo period (16 light hours), the values of r^2^ were 0.78, 0.84 and 0.03 where there was a better ratio in the synthesis of lipophilic and hydrophilic metabolites in sodium acetate as a substrate at a concentration of 0.001 g/L under a long photoperiod (16 h of light).

Glycerol, after a photoperiod of 8 h of light (short) with *Pearson’s correlation* had an r^2^ value of 0.12, 0.04 and 0.06; therefore, there was no correlation, under this condition, in the synthesis of lipophilic and hydrophilic metabolites. Regarding the photoperiod of 16 h of light (long), the value of r^2^ was 0.76, 0.13 and 0.03 for the concentrations of 0.27, 0.33 and 0.44 g/L, respectively; therefore, only the first condition was favorable, and induced a slightly linear synthesis of both lipophilic and hydrophilic antioxidant metabolites. With the previous analyses taken into account, our preliminary conclusion was that the condition where the measured response variables were favored was that of sodium acetate at photoperiods of 16 h of light and 8 h of darkness, under a concentration of 0.001 g/L or 0.0005 g/L.

With the quantified metabolites, cell growth and biomass concentration, the kinetic characterization of *C. vulgaris* with glycerol and sodium acetate as substrates was possible; it was found that, with sodium acetate as a substrate, *C. vulgaris* could grow adequately in a concentration of 0.001 g/L, in a photoperiod of 16 h of light and 8 h of darkness, with a doubling time of 4.84 h (Td) and a division speed σ = 0.20 h^−1^, having a biomass concentration at the end of the kinetics of 2.075 g/L at hour 240. Regarding glycerol as a substrate, it was found that, at a concentration of 0.33 g/L of glycerol, and after the same photoperiod (16 h of light and 8 h of darkness), the doubling time (Td) was 4.63 h, with a maximum division rate of σ = 0.18 h^−1^, and with a biomass concentration at the end of the kinetics of 1.4 g/L. However, even though the doubling time was less than that observed with sodium acetate as a substrate, as well as the amount of metabolites with antioxidant activity, the best substrate for the synthesis of these metabolites and for cell growth was sodium acetate under prolonged photoperiods of 16 h of light and 8 h of darkness.

Finally, the specific growth rates were calculated; the calculation was carried out with the initial concentration of biomass on day 0, and the day on which the maximum concentration was obtained, using the following Equation (1) [23]:
(1)$$\mu =\ln(\frac{\frac{W2}{W1}}{t2-t1})$$
where *W*1 and *W*2 were the biomass concentration (g/L), and *t*1 and *t*2 were the time in which the change in biomass concentration, expressed in days, was evaluated. With this, the results for a photoperiod of 16 h of light and 8 h of darkness, with sodium acetate as substrate, were 0.0425 g L^−1^ day^−1^, 0.5365 g L^−1^ day^−1^ and 0.2682 g L^−1^ day^−1^. Sodium acetate, at a concentration of 0.001 g/L, was therefore the best condition for obtaining biomass. Regarding glycerol under the same photoperiod, a rate of 0.1823 g L^−1^ day^−1^ was obtained for a concentration of 0.44 g/L; for the rest of the conditions, values equal to zero or negative were obtained since, as can be seen In Figure 2, the behavior of the biomass was variable throughout the 240 h, despite the fact that the samples were homogenized manually and electrically.

## 3. Discussion

Comparing a study in 2018 with the above results, it can be observed that the exponential phase began at hour 48, as reported in this study, reaching the stationary phase around hour 144, so that the behavior was similar, which meant that *C. vulgaris* was growing favorably when using sodium acetate or glycerol as a substrate. According to the report by Li, sodium carbonate was used at a concentration of 160 mM, and the conditions of sodium acetate for this worked as well as glycerol—equivalences were 6.1 mM, 12.2 mM and 18.3 mM for each substrate; for this reason, the presence of sodium ions was not the cause of *C. vulgaris* being inhibited, because it is not a halophile organism that maintains sodium ions for cell maintenance [27].

Even when allowing for the above, it can be observed that the condition of sodium acetate after a long photoperiod was favorable for the formation of biomass. The biomass generated was favorable by comparison with results of a study by Heredia-Arroyo et al. 2011, in which, under autotrophic, heterotrophic and mixotrophic growth at 160 rpm at 30 °C, and with glucose as a substrate at a concentration of 4 g/L, biomass was 0.4 g/L, 0.75 g/L and 1.40 g/L, respectively. The maximum points obtained in this study, however, were 3.8 g/L for the sodium acetate condition at concentration of 0.0015 g/L of substrate at hour 24, and at hour 168, 2.4 g/L was obtained for the same condition, followed by the 0.44 g/L glycerol condition with 1.8 g/L biomass concentration at hour 168; both conditions were in the long photoperiods (16 h of light and 8 h of darkness), so that these results reaffirmed that the kind of photoperiod has a significant effect on the biomass generated and on the medium formulation—the strain and the operational conditions at the flask level allowing higher yields than those already worked, so that replicating these conditions at photobioreactor level in future research could achieve greater results; moreover, the use of less substrate could save resources, compared to the use of glucose in a conventional way [28].

In the year 2013, the growth of *C. vulgaris* in wastewater was evaluated, adding one sodium acetate condition and another glycerol condition as substrate. In the sodium acetate, concentrations were variated in 5, 10 and 15 mM, and the results were a maximum biomass concentration of 1.96 g/L, 2.66 g/L and 3.4 g/L; by comparison, in this study, at sodium acetate concentrations of 6.1 mM, 12.2 mM and 18.3 mM, the results were a maximum biomass concentration of 2 g/L, 3 g/L and 3.6 g/L; the preparation of the medium had a significant effect on production, compared to that previously reported, as sodium nitrates were present in the residual water components, favoring growth, and so this study was able to generate better yields, since only sodium acetate was available as a source of metabolism [29].

Recently, in 2019, the growth of *C. vulgaris* was evaluated under stress, in a photoheterotrophic medium, with different concentrations of sodium acetate (1 g/L, 2 g/L and 4 g/L)—thus, as sodium acetate added with iron sulfate at the same acetate concentrations of 1 g/L, but varying the iron sulfate concentration at 0.5 g/L and 1 g/L. The results showed that sodium acetate only had a maximum biomass concentration of 2.2 g/L for the condition of 1 g/L of sodium acetate, and a concentration of 3.5 g/L in the medium with the addition of sulfate of iron of 0.5 g/L. As can be seen, the results were close to those obtained in this study; however, the substrate used was better, as this research used concentrations of 0.005 g/L, 0.001 g/L and 0.0015 g/L, generating savings of inputs, where the operational conditions gave higher yields for technological application. This variation may have been due to the time of exposure to light as, in a study by El-Sheekh et al., the photoperiod was not specified [30].

As can be observed, the kind of photoperiod is a factor that has great significance for microalgae growth, as a longer time of exposure to radiation allows for greater efficiency in photosynthesis and generates a greater number of cells that are not proportional to the biomass synthesized. To support the above, we can cite a study by Blair et al. 2014, where the type of light was evaluated in growth kinetics at flask level in 200 mL, 25 °C and 160 rpm shaking, similar to that experienced in this study. Blair et al. observed that, of the analyzed light colors (red, blue, white and green), the one with the best biomass yield was where white light was used followed by blue light, with a concentration of 0.04 g/L per day and 0.02 g/L per day, compared with what was obtained for sodium acetate and glycerol at 16 h of light and 8 h of darkness, with a concentration of 0.38 g/L per day and 0.15 g/L per day; the conditions experienced were favorable, and the red LED-type light used allowed the absorption of light to optimize the generation of biomass. In addition, the same study’s observations of the behavior of biomass were similar to our own study; there was no exponential kinetic trace, but it was more linear, and cell division had a similar response under white light, by which it was proven that the type of light had an effect on cell division, by generating larger cells, but that it did not generate the proportional amount of biomass that would have generated a similar trace in both parameters [19]. On the other hand, in regard to the chemical composition of *C. vulgaris*, depending on the kind of light to which it was exposed, we note the study carried out by Kula et al. in 2013, where the content of some metabolites was evaluated in fluorescent light, being exposed to blue and red light and to red light in the far red range; this was carried out by means of LED light over 10 days, and it was observed that the best growth occurred in red and blue light, and that this was where metabolic activity was favored. The metabolic activity was measured through the emission of heat energy in the cultures associated with the processes of intensity of cellular respiration. In addition, it was observed that microalgae, when exposed to these radiation conditions, generated metabolites that allowed for regulating the effect of light absorption, so that it did not affect them. These molecules could, by Raman spectroscopy, indicate terpenes, because of the double band to convert them into chromophores molecules that absorbed light and thus prevented them from damaging other cellular components. Among these compounds, we noted the β-carotenes and even the polyphenols, which were known to have antioxidant activity, which is why this shows that the response in the stabilization of the synthetic radicals of ABTS•+ and DPPH• was due to the presence of these molecules, as both methods measured the activity of fat-soluble molecules such as carotenes (DPPH•) and water-soluble molecules such as polyphenols (ABTS•+) [31].

The specific rates of biomass (µ) were compared, and it was found that, in a photoheterotrophic medium with 2 and 10 g/L of substrate, for glucose there was a value of 0.390 g L^−1^ day^−1^ and 0.475 g L^−1^ day^−1^, xylose of 0.173 g L^−1^ day^−1^ and 0.020 g L^−1^ day^−1^, sucrose 0.176 g L^−1^ day^−1^ and 0.056 g L^−1^ day^−1^, maltose 0.447 g L^−1^ day^−1^ and 0.482 g L^−1^ day^−1^, sodium acetate 0.430 g L^−1^ day^−1^ and 0.456 g L^−1^ day^−1^, glycerol 0.160 g L^−1^ day^−1^ and 0.172 g L^−1^ day^−1^, which were lower in some cases than those reported in this study. The sodium acetate condition after 16 h of light, and with a concentration of 0.001 g/L of substrate, had a value of 0.5365 g L^−1^ day^−1^, which was higher than those reported by Kong et al. Similarly, for the glycerol condition at 16 h of light and with a substrate concentration of 0.44 g/L, the result was higher; therefore, it can be concluded that the conditions evaluated were significant for biomass synthesis [23]. Due to the behavior of the trace in Figure 2, however, a more robust analysis is recommended.

## 4. Materials and Methods

Cell growth. The strain of *C. vulgaris* CIB45 used was obtained from the Centro de Investigaciones Biológicas del Noroeste S.C. La Paz Baja California Sur Mexico; it was maintained in the Biochemical Laboratory of the Tecnológico Nacional de México campus Morelia, under 3N-BBM+V medium. The maintenance medium specifications were NaNO_3_ (8.82 mM), CaCl_2_·2H_2_O (0.17 mM), MgSO_4_·7H_2_O (0.3 mM), K_2_HPO_4_·3H_2_O (0.43 mM), KH_2_PO_4_ (1.29 mM), NaCl (0.43 mM), stock solution of P-IV [Na_2_EDTA (2 mM), FeCl_3_·6H_2_O (0.36 mM), MnCl_2_·4H_2_O (0.21 mM), ZnCl_2_·6H_2_O (0.037 mM), CoCl_2_·6H_2_O (0.0084 mM) and Na_2_MoO_4_·2H_2_O (0.017 mM)]. The maintenance of the cell was carried out with an initial cell culture of 1.0^7^ cells/mL, in a photoperiod of 16 h of light and 8 h darkness, in closed glass flasks, and without shaking. The maintenance medium was a 200 mL final volume in a 250 mL capacity Erlenmeyer flask. The photoperiod was selected because, in explorative runs, those were the best conditions in which to form new cells. The temperature for growing was from 30 °C [1,32]. Finally, the type of light used was red LED, because it was observed that the use of red light promoted the generation of cells under uniform divisions, and the generation of similar cell sizes [27]. The light intensity worked was 405 luxes.

Kinetics growth. Two organic carbon conditions were established, being sodium acetate and glycerol at different concentrations (0.05%, 0.1% and 0.15% to each one *v*/*v*). To calculate the kinetic parameters, the concentrations were expressed in g/L; for sodium acetate, the equivalents were 0.0005 g/L, 0.001 g/L and 0.0015 g/L, and for glycerol, the equivalents were 0.27 g/L, 0.33 g/L and 0.44 g/L, with respect to their molecular weight and under different times of exposure to light (photoperiod); the first photoperiod was 8 h of light and 16 h of darkness, and the second was 16 h of light and 8 h of darkness. The kind of LED light was red. The growth kinetics were started at a cell concentration of 5.0^7^ cells/mL; for sodium acetate, growth kinetics were performed at 10 days, with a sample taken every 24 h at an operating temperature of 30 °C and with a stirring of 150 rpm.

### 4.1. Response Variables

Cell growth. A microscope count was performed, of the formation of new cells during the kinetics, with a Neubauer camera at No. 40 objective; it was not necessary to perform a methylene blue staining of the cells, because viable cells were clearly distinguishable, being those that kept their cell wall intact [33].

Biomass. For the biomass technique, dry weight was used. Under aluminum trays at constant weight, 1 mL of sample was placed, and was subjected to heat in an electric oven for 24 h at a temperature of 60 °C, after which, the tray with the sample was re-weighed and, by the difference in weights, the number of cells per liter of culture medium was calculated [34].

Substrate consumption. The calculation of the consumed equivalents of sodium acetate was carried out under a potentiometric titration technique, involving the construction of a calibration curve with NaOH (1 M) and HCl (5%). To make the calibration curve, different solutions of the stock solution of NaOH (1 M) were made: 0.1; 0.2; 0.3; 0.4; 0.5; 0.6; 0.7; 0.8; 0.9; and 1 M. These dilutions were neutralized with the stock solution of HCl (5%), and the volume required to reach the neutralization of the medium (pH 7) was recorded [35]. Once the readings were obtained, the calibration curve was built and, through a *Pearson’s correlation* with a value of r^2^ of 0.99, the equation of the linear behavior of the neutralization was established. With *Pearson’s correlation* and using a volume of 5 mL of sample with the *C. vulgaris* growth medium, the required volume expenditure of HCl was recorded, to neutralize the medium, and to thus indirectly calculate the concentration of acetate consumed from the first day of cell growth [36]. With regard to glycerol, an enzymatic technique was carried out, based on the phosphorylation of glycerol by adenosine triphosphate (ATP), to obtain glycerol-3-phosphate; this with the help of the enzyme glycerol kinase. Subsequently, the glycerol-3-phosphate was converted to dihydroxyacetone phosphate—catalyzed by the reaction of the enzyme glycerol-3-phosphate dehydrogenase—to generate nicotinamide and adenine dinucleotide reduced (NADH). The increase in NADH concentration was measured by the change in absorbance at 340 nm, which was proportional stoichiometrically to the increase in glycerol concentration. The process which followed was the preparation of the following stock solutions: MgCl_2_ (1 M); glycine (0.5 M); hydrazine (1.5 M); ATP (50 mM); β-NAD (20 mM); glycerol kinase (85 Ku/L); glycerol-3-phosphate dehydrogenase (1700 U/L); glycerol solution (10 mM); perchloric acid (0.5 M); and KOH (10 M). The reagents were added in the order in which they appear in the previous paragraph, in such a way that when the glycerol kinase was added, the reading was recorded in the UV-vis spectrophotometer at 340 nm; it was continued with the addition of the rest of the solutions, and when the enzyme glycerol-3-phosphate dehydrogenase was added, a second reading was made under the same wavelength, after 10 min of the reaction of the reagents. With this, a calibration curve was constructed, to calculate the glycerol concentration in the kinetic samples of *C. vulgaris* [35].

Antioxidant activity. For the quantification of the products formed, the ABTS•+ and DPPH• technique was carried out, which indirectly measured the concentration of polyphenolic compounds with said activity. The metabolites detected corresponded to polyphenols, as the ABTS•+ technique measured the stability of the ABTS•+ radical by hydrophilic molecules and, within the metabolites synthesized by autotrophic beings, polyphenolics have a hydrophilic character, due to hydroxyl groups in the multiple substitutions in their aromatic rings. Regarding DPPH•, it is a technique in which the synthetic radical is stabilized by lipophilic molecules, such as terpenoids or alkalis. It has been reported, however, that *C. vulgaris*, through a profile of analyzed pigments, has the ability to synthesize terpenoids and polyphenols; that is why, with the DPPH• technique, it was possible to quantify terpenoid metabolites with antioxidant activity that synthesized microalgae [37]. For this, cell rupture was carried out with glass beads, by carrying out previous washes with deionized water under centrifuge at 4000 rpm for 5 min. This operation was run three times, to eliminate the salts dissolved in the medium. The volume of deionized water used in the wash was 1 mL. Once the cells had been washed, they were put into contact with the glass beads suspended in 1 mL of deionized water in a centrifuge at 4000 rpm, to later take the supernatant and evaluate the stability of the synthetic radicals of ABTS•+ and DPPH•. Regarding the ABTS•+ method, a radical ABTS•+ solution was prepared at 7 mM concentration, ammonium persulfate aqueous solution at 2.5 mM concentration and a Trolox stock at 80 µM/mL concentration. Subsequently, a calibration curve was made in Trolox equivalents (µM/mL) under a 734 nm wavelength. Once the calibration curve was constructed, the inhibition percentage of the ABTS•+ radical solution, when in contact with the Trolox solution, was calculated, and the concentration in Trolox equivalents was calculated too. For the sample activity determination, a volume of 10 μL of sample was diluted with 990 μL of adjusted ABTS•+ solution, and the corresponding readings and calculations were carried out to express the results in equivalents of Trolox g/L [37].

In the DPPH• method, a radical DPPH• solution was prepared at 0.046 g/L concentration, and a Trolox stock at 80 µM/mL concentration. Subsequently, a calibration curve was made in Trolox equivalents (µM/mL) under a 515 nm wavelength. Once the calibration curve was constructed, the inhibition percentage of the DPPH• radical solution, when in contact with the Trolox solution, was calculated, and the concentration in Trolox equivalents was calculated too. For the sample activity determination, a volume of 100 μL of sample was diluted with 3900 μL of adjusted DPPH• solution, and the corresponding readings and calculations were carried out to express the results in equivalents of Trolox g/L [37].

### 4.2. Statistic Design

For the statistical analysis, a “screening experimental design (SED)” was run. With an ANOVA analysis, the data were compared, through Origin 6.0 Professional, JMP 6 and Statgraphics Centurion software, with a significance value of 0.05. The SED was run, taking two independent factors and five dependent variables. The independent factors evaluated were type of substrate (sodium acetate and glycerol—varying their concentration—and type of photoperiod, varying the time of exposure to light: the first case was 8 h of light and 16 h of darkness; the second case was the inverse—16 h of light and 8 h of darkness. The dependent variables analyzed were: number of cells formed; biomass; substrate consumption; and antioxidant activity of secondary metabolites by ABTS•+ and DPPH• method. The number of experimental runs for each substrate at flask level was 6, under the different substrate concentrations and the different photoperiods; in addition to this, the design was carried out in triplicate.

## 5. Conclusions

With the study carried out, it can be concluded that *C. vulgaris* is a species of microalgae that can grow favorably at flask level under photoheterotrophic growth, with sodium acetate or glycerol as substrate. Its growth, however, is favored when working with sodium acetate, as a greater formation of cells, biomass and formation of products with antioxidant activity was observed. In addition, the concentration of the substrate had a significant effect on the kinetics of *C. vulgaris*, since at high concentrations growth can be inhibited specifically in mix 3N-BBM+V and glycerol medium, which can be either by saturation and interference in the metabolic chains or by preventing the substrate from being available to the microalgae. On the other hand, it can be concluded that the type of photoperiod has a significant effect on the growth and synthesis of metabolites with antioxidant activity, as there was greater activity in long photoperiods. Finally, it can be concluded that *C. vulgaris* is capable of synthesizing metabolites with antioxidant activity and storing them inside the cell as protection against stress, and that these abilities could be exploited at the biotechnological level.

## Figures and Tables

**Figure 1 molecules-27-06346-f001:**
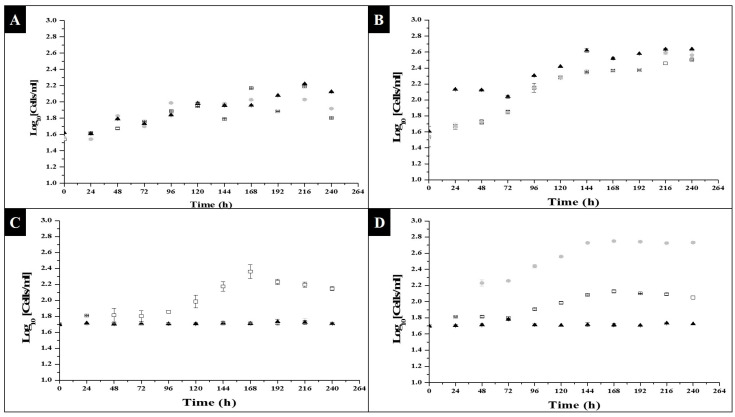
Cell growth of *C. vulgaris* under photoheterotrophic medium: (**A**) sodium acetate in 8 light hours and 16 darkness photoperiods; (**B**) sodium acetate in 16 light hours and 8 darkness photoperiods. □0.0005 g/L, ●0.001 g/L and ▲0.0015 g/L sodium acetate concentration; (**C**) glycerol in 8 light hours and 16 darkness photoperiods; (**D**) glycerol in 16 light hours and 8 darkness photoperiods. □0.27 g/L, ●0.33 g/L and ▲0.44 g/L glycerol concentration.

**Figure 2 molecules-27-06346-f002:**
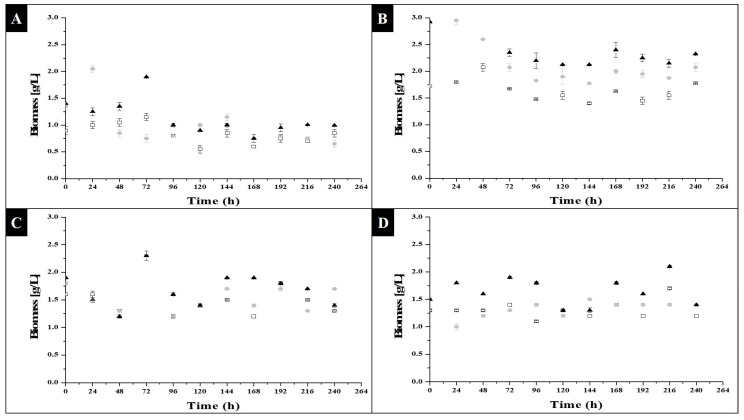
Biomass of *C. vulgaris* under photoheterotrophic growth: (**A**) sodium acetate in 8 light hours and 16 darkness photoperiods; (**B**) sodium acetate in 16 light hours and 8 darkness photoperiods. □0.0005 g/L, ●0.001 g/L and ▲0.0015 g/L sodium acetate concentration; (**C**) glycerol in 8 light hours and 16 darkness photoperiods; (**D**) glycerol in 16 light hours and 8 darkness photoperiods. □0.27 g/L, ●0.33 g/L and ▲0.44 g/L glycerol concentration.

**Figure 3 molecules-27-06346-f003:**
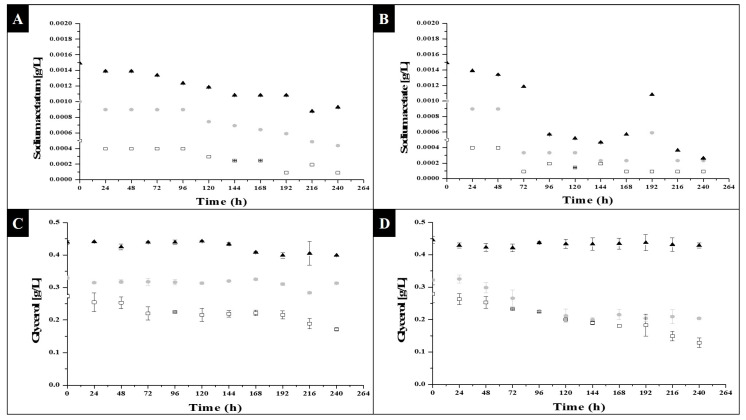
Substrate consumption of *C. vulgaris* under photoheterotrophic growth: (**A**) sodium acetate in 8 light hours and 16 darkness photoperiods; (**B**) sodium acetate in 16 light hours and 8 darkness photoperiods. □0.0005 g/L, ●0.001 g/L and ▲0.0015 g/L sodium acetate concentration; (**C**) glycerol in 8 light hours and 16 darkness photoperiods; (**D**) glycerol in 16 light hours and 8 darkness photoperiods. □0.27 g/L, ●0.33 g/L and ▲0.44 g/L glycerol concentration.

**Figure 4 molecules-27-06346-f004:**
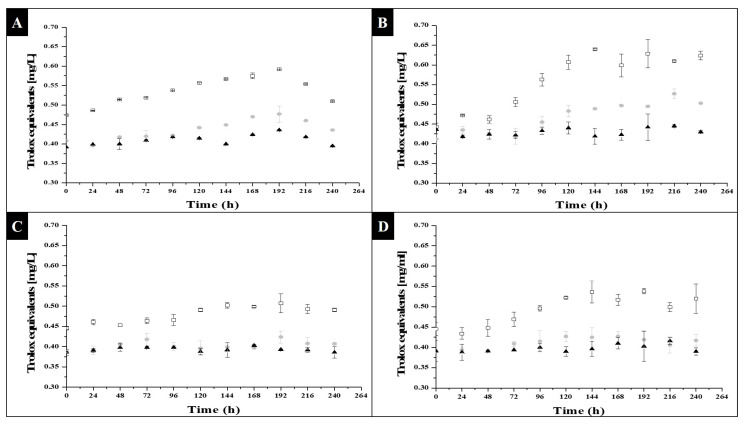
Antioxidant activity of *C. vulgaris* under photoheterotrophic growth with ABTS•+ method: (**A**) sodium acetate in 8 light hours and 16 darkness photoperiods; (**B**) sodium acetate in 16 light hours and 8 darkness photoperiods. □0.0005 g/L, ●0.001 g/L and ▲0.0015 g/L sodium acetate concentration; (**C**) glycerol in 8 light hours and 16 darkness photoperiods; (**D**) glycerol in 16 light hours and 8 darkness photoperiods. □0.27 g/L, ●0.33 g/L and ▲0.44 g/L glycerol concentration.

**Figure 5 molecules-27-06346-f005:**
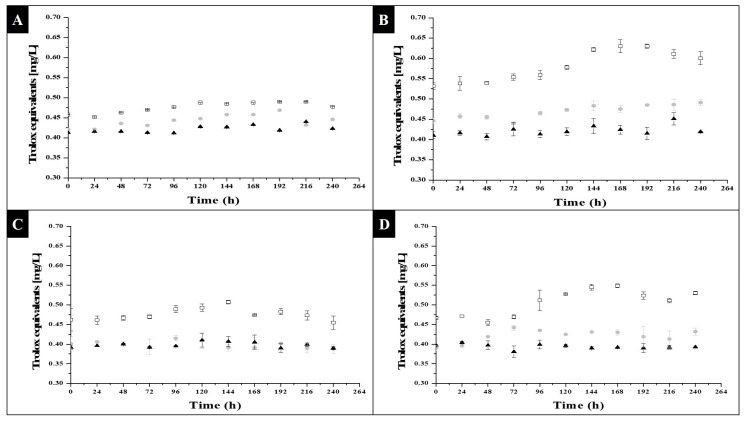
Antioxidant activity of *C. vulgaris* under photoheterotrophic growth with DPPH• method: (**A**) sodium acetate in 8 light hours and 16 darkness photoperiods; (**B**) sodium acetate in 16 light hours and 8 darkness photoperiods. □0.0005 g/L, ●0.001 g/L and ▲0.0015 g/L sodium acetate concentration; (**C**) glycerol in 8 light hours and 16 darkness photoperiods; (**D**) glycerol in 16 light hours and 8 darkness photoperiods. □0.27 g/L, ●0.33 g/L and ▲0.44 g/L glycerol concentration.

## Data Availability

Not applicable.

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
