# Peer review of "Antioxidant Activity and Kinetic Characterization of Chlorella vulgaris Growth under Flask-Level Photoheterotrophic Growth Conditions"

_molecules, 2022, doi:10.3390/molecules27196346_

Round 1

Reviewer 1 Report

Jesús Alberto Coronado-Reyes et al. studied the growth of C. vulgaris under photoheterotrophic conditions at the flask level.

You will find the most important remarks highlighted in yellow. The writing of the document should be more careful and several points need to be really improved.

Best regards,

Author Response

Observations

Point 1: The writing of the document should be more careful and several points need to be improved.

Response 1: In the entire manuscript a lot of sentences should be shortened (add full stops). This was the observation/ Correction: We review and add the necessary points in all text.

Point 2: Example n°1 line 42 to 47: «As mentioned above, C. vulagris has a large field of research and application regarding the primary metabolites that it synthesizes. However, very little has been reported on secondary metabolites, such as polyphenolic or terpene compounds, that have antioxidant activity. So research the

presence and optimal production of these molecules with...»

Response 2: Line 42 to 47: The sentence was changed to «As mentioned above, C. vulagris has a large field of research and application regarding the primary metabolites that it synthesizes. However, very little has been reported on secondary metabolites, such as polyphenolic or terpene compounds, that have antioxidant activity. So research the presence and optimal production of these molecules with...».

Point 3: Example n°2 line 47 to 53: «Talking about of primary metabolism, there is some research on C. vulgaris for the accumulation of lipids based on organic residues in sewage sludge (DIG-OFMSW) as well as in growth media in wine residues (DIG-WL). It has

Response 3: Line 47 to 53: The sentence was changed to «Talking about of primary metabolism, there is some research on C. vulgaris for the accumulation of lipids based on organic residues in sewage sludge (DIG-OFMSW) as well as in growth media in wine residues (DIG-WL). It has been observed».

Point 4: Example n°3 line 57-64: «If ideal or majority concentrations of nitrogen are found in the medium, the microalgae will have greater protein synthesis. However, research has been carried out where nitrogen used. It is for them that in 2019 Parra et al., evaluated the culture growth, microalga biomass productivity and protein yield»

Response 4: Line 57 to 64: The sentence was changed to «If ideal or majority concentrations of nitrogen are found in the medium, the microalgae will have greater protein synthesis. However, research has been carried out where nitrogen used. It is for them that in 2019 Parra et al., evaluated the culture growth, microalga biomass productivity and protein yield».

Point 5: Example n°4 line 69-76: Regarding the content of carbohydrates, in 20

of Glycerol and Glucose on the Improvement of Production of Biomass, Lipids and Soluble Carbohydrates by Chlorella vulgaris in mixotrophic that glycerol as the only source of substrate does not increase the yields of the expected products. However, the synergy with glucose increases the yields of ».

Response 5: Line 69 to 76: the sentence was changed to «Regarding the content of carbohydrates, in 20 of Glycerol and Glucose on the Improvement of Production of Biomass, Lipids and Soluble Carbohydrates by Chlorella vulgaris in mixotrophic that glycerol as the only source of substrate does not increase the yields of the expected products. However, the synergy with glucose increases the yields of ».

Point 6:

+ Line 243 + Line

- In the whole document ml mL

Response 6: Check the spaces/ Correction: the spaces and points were checked.

Homogenise the positions of the spaces, especially for the unites. / Correction: the position of the spaces in unites were homogenized (M, %, g/L, mL, mM).

In the whole document ml -> mL/ Correction: in the document ml was changed to mL.

Point 7: ABSTRACT

- Line 19: 1,825 1.825 g/L

- Line 19: 0.20 h-1 : I can't find this data in the article

Response 7:

  1. Line 19: 1,825 -> 1.825 g/L. / Correction: in the line, 1,825 was changed to 1.825 g/L.
  2. Line 19: 0.20 h-1: I can't find this data in the article/ Correction: line 19: 0.20 h-1 was add in “Results” section.

Point 8: II. INTRODUCTION

- Line 43: «However, very little has been reported on secondary metabolites, such as polyphenolic or terpene

compounds, that have antioxidant activity. » Can you tell us about the state of knowledge on these secondary

metabolites? Your introduction talks about primary metabolites but this is not the subject of this article.

- There is no information on sodium acetate and glycerol in the introduction.

Why did you choose these 2 organic carbon conditions and how did you choose the concentrations tested?

- You need to talk about the use of red light here. I think this is important. Can you provide the spectrum? and

the intensity used (PAR and PUR)?

- Line 42: C. vulagris C. vulgaris

- Line 47: go to the line before «

- Line 51: I think there are some words missing: «a biomass concentration of 1.36 ± 0.09 g/L was obtained for

the DIG-WL medium and 1.05 ± 0.13 g/L for the DIG-OFMSW medium»

- Line 51: According to article [35], there is a mistake: «having a lipid accumulation of 28.86 ± 0.05% for the

DIG-WL medium and 6.1 ± 0.2% for the DIG-OFMSW medium. »

- Line 61: microalga microalgae

- Line 62: Chlorella vulagris C. vulgaris

- Line 81 to 83: In my opinion this sentence is not clear: «it was observed that there was a decrease in the

concentrations of chlorophylls (Chl a and Chl b), decrease in new cells formed and concentration of

carotenoids decreasing by 54% and 93%, 60% and 74%, 50% and 72% respectively, with the synthesis of

proteins and the antioxidant defense system ».

I suggest you rewrite it. I don't understand what the percentages mean, especially as they are increasing.

- Line 85: C. vulgaris in italics

Response 8:

Line 43: «However, very little has been reported on secondary metabolites, such as polyphenolic or terpenecompounds, that have antioxidant activity.» Can you tell us about the state of knowledge on these secondary metabolites? Your introduction talks about primary metabolites but this is not the subject of this article/ Correction: we talking about the state of knowledge on these secondary metabolites.

  1. There is no information on sodium acetate and glycerol in the introduction. Why did you choose these 2 organic carbon conditions and how did you choose the concentrations tested?/ Correction: We talk about works about glycerol and sodium acetate like substrate to C. vulgaris and we selected the concentrations for this work since, with what was observed in the work of Kong et al., a concentration of 2 and 10 g/L produced an inhibition in the synthesis of secondary metabolites (pigments) but their synthesis was favored under conditions of stress with respect to the substrate, and since the objective of the work is to promote the synthesis of metabolites with antioxidant activity, it was decided to work with lower concentrations. In addition, decreasing concentrations starting from 1 g/L were tested and the conditions reported were those that, under a screening experiment design, were favored for both substrates.
  2. You need to talk about the use of red light here. I think this is important. Can you provide the spectrum? and the intensity used (PAR and PUR)?/ Correction: we talk and explane the importance about red light.
  3. Line 42: C. vulagris -> C. vulgaris/ Correction: in line 42: C. vulagris was changed to C. vulgaris.
  4. Line 51: I think there are some words missing: «a biomass concentration of 1.36 ± 0.09 g/L was obtained for the DIG-WL medium and 1.05 ± 0.13 g/L for the DIG-OFMSW medium» / Correction: in line 51: The sentence was changed to «a biomass concentration of 1.36 ± 0.09 g/L was obtained for the DIG-WL medium and 1.05 ± 0.13 g/L for the DIG-OFMSW medium».
  5. Line 51: According to article [35], there is a mistake: «having a lipid accumulation of 28.86 ± 0.05% for the DIG-WL medium and 6.1 ± 0.2% for the DIG-OFMSW medium.»/ Correction: in line 51: the mistake was corrected to «having a lipid accumulation of 28.86 ± 0.05% for the DIG-WL medium and 6.1 ± 0.2% for the DIG-OFMSW medium.». However, the remark in line 51 talks about primary metabolites and was modified to remove it to emphasize secondary metabolites.
  6. Line 61: microalga -> microalgae/ Correction: in line 61: The microalga word was changed to microalgae.
  7. Line 62: Chlorella vulagris -> C. vulgaris/ Correction: in line 62: the Chlorella vulagris word was changed to C. vulgaris.
  8. Line 81 to 83: In my opinion this sentence is not clear: «it was observed that there was a decrease in the concentrations of chlorophylls (Chl a and Chl b), decrease in new cells formed and concentration of carotenoids decreasing by 54% and 93%, 60% and 74%, 50% and 72% respectively, with the synthesis of proteins and the antioxidant defense system ». I suggest you rewrite it. I don't understand what the percentages mean, especially as they are increasing./ Correction: we deleted this information.
  9. Line 85: C. vulgaris in italics/ Correction: in line 85: C. vulgaris was changed to italics style. The word was changed and cheked in all the document.

Point 9: II. RESULTS

- Line 91: 50 x 10 6 cells / ml 5.107 cells/mL

- Line 93 and 102: C. vulgaris in italics

- I'm sorry I don't understand, are we looking at cell growth or cell division in Figure 1?

- Line 108 : can you explain your r2 please? Why don't you talk about p-value?

Figure 1 A/B

- This paragraph is very confusing for me.

- I am surprised by the low number of cells/mL, especially if the initial concentration is 5.107 cell/mL. As it

is, I understand 50 cells/mL. Can you explain it to me.

- Why do you use a high initial concentration? I think this is why the 3 phases of growth (latency, expo and

stationary) are not clearly identifiable.

- For all experiments in Figure 1, it would be interesting to have the calculation of the growth rates.

Figure 1A: Unless I am mistaken, the highest value was obtained at hour 216. The latency phase, the

exponential phase and the stationary phase are difficult to identify on this graph... It would be preferable to

represent the y-axis in linear and not in log10 for Figure 1 (A to D). Interpretation should be easier.

Moreover, there is a problem with values at hour 144 (an explanation?). You need to discuss the problem at

hour 144. Perhaps this experiment should be repeated.

Figure 1B: I don't totally agree with your observation of the latency phase, the exponential phase and the

stationary phase. The highest value was obtained at hour 144 or 216 (?). In the text, line 110, there is a problem

with values, especially with the value 1.71.

The interpretation of figure 1A-B needs to be reviewed.

Figure 1 C/D

- Line 134 to 138: There is probably an inversion « With the Tukey-Kramer HDS analysis, it was possible to

identify that the glycerol conditions at a concentration of 0.33 and 0.44 g/L at a SHORT photoperiod (8

hours of light) and the condition of 0.44 g/L at a PROLONGED photoperiod (16 hours of light), are not

significantly different since there was no cell division. However, the condition that favors growth and division

is that of 0.33 g/L of glycerol in prolonged photoperiods.

- Line 138: « glycerol in prolonged photoperiods. As well as it can be seen »

FIGURE 2

- Line 151: To overcome this problem, the biovolum (μm3/mL) should be obtained using a particle counter,

especially for this microalgae OR you can do the same protocol (dry biomass) but with a higher volume of

culture (ex 20 mL).

I think the initial concentration is too high to see a real impact on the growth so on the biomass. Unless I am

mistaken, you start with 5.107 cells/ mL and you have 2.108 cells/mL at the end of the experiment i.e. only 2

doublings of the population.

- Line 155: It is imperative to make the culture as homogeneous as possible before sampling for analysis.

Homogenisation can be done manually.

- Line 165: «0.0015 g/L of substrate, respectively.»

- Why you don't mention figure 2C and D (glycerol resultas)?

FIGURE 3

- Line 182 to 187: here is my proposal: «As can be seen in Figure 3A and 3B there was a significant

consumption in the different concentrations of substrate. In figure 3B a greater decrease can be seen. This

behavior is in agreement to that observed in Figure 1B, where it was the experimental condition with a

greater formation of cells. In figure 3A the decrease was less pronounced, as well as the minor increases in

cells of Figure 1A.»

- Line 193 and 199: deletes a point

- Line 196 to 199 : « in Figure 1C. » The rest of the sentence, please rewrite.

FIGURE 4/5

- Line 225: that This that this

- Line 256 : 0.001g/L, are you sure?

Response 9:

  1. Line 91: 50 x 10 6 cells/ml-> 5.107 cells/mL/ Correction: in line 91, 50x106 cells/ml was changed to 5.107 cells/mL.
  2. Line 93 and 102: C. vulgaris in italics/ Correction: in line 93 and 102: C. vulgaris was changed to italics style.
  3. I'm sorry I don't understand, are we looking at cell growth or cell division in Figure 1? It´s the same?/ Correction: in response we change the words cell division to cell growth because is diferente.
  4. Line 108 : can you explain your r2 please? Why don't you talk about p-value?/ Correction: we decided to place the values of the r2 because this parameter reflects the linearization or dependence of one response with respect to another, that is, if a response such as cell growth is varied, it is possible to observe if the antioxidant activity is proportional to this change, having this response a dependency or otherwise, the variation in cell growth does not reflect or have an effect on antioxidant activity. P values were not included because they only show us the significant difference between conditions, as mentioned in some results, since we are interested in the dependency relationship between the measured responses.

Figure 1 A/B

  1. This paragraph is very confusing for me/ Correction: we think is necessary this description and we change the references about it to be clear.
  2. I am surprised by the low number of cells/mL, especially if the initial concentration is 5.107 cell/mL. As it is, I understand 50 cells/mL. Can you explain it to me/ Correction: the inicial number of cells is 50,000000 cell/mL. 500 µL sample was taken and placed in a Neubauer chamber, observing with objective No. 40 and counting in the 5 quadrants (upper left, upper right, lower right, lower left and central). By probability of counting in post triplicate samples, it is indicated that the registered number must be multiplied by a value of 1x106 to obtain an estimated concentration of cells/mL
  3. Why do you use a high initial concentration? I think this is why the 3 phases of growth (latency, expo and stationary) are not clearly identifiable/ Correction: for the kinetics, it was decided to start with a concentration of 5.07 cells/mL because in previous exploratory studies, the kinetics were started with concentrations of 1.07, 2.07 and 3.07 cells/mL, but at the time of sampling, no significant amount was observed. considerable number of cells in the Neubauer chamber, so the first sample takings indicated a lower concentration than the initial ones and there was too much variation in the cell count. With a concentration of 5.07 cells/mL this count was better observed. Regarding the behavior in the kinetic phases, it is possible to appreciate them under the conditions of Figure 1B and 1D since the effect of the photoperiod is observed. However, it is believed that there is not a greater number of cells because the inorganic substrate is limiting and this is reaffirmed with Figure 3. For this work, optimization conditions are sought, so a low amount of substrate but enough to promote cell division as well as promote the synthesis of metabolites due to the stress condition.
  4. The rates were calculated.
  5. Figure 1A: Unless I am mistaken, the highest value was obtained at hour 216. The latency phase, the exponential phase and the stationary phase are difficult to identify on this graph... It would be preferable to represent the y-axis in linear and not in log10 for Figure 1 (A to D). Interpretation should be easier. Moreover, there is a problem with values at hour 144 (an explanation?). You need to discuss the problem at hour 144. Perhaps this experiment should be repeated/ Correction: it was decided to place the Log10 since it was generally found in this way in the consulted references. In addition, for the calculations in the kinetic characterization, this value is necessary and that is why it was decided to represent it in this way. Regarding the behavior of Figure 1A, the values behavior is because we detected a mistak about 0.001 and 0.0015 g/Lconcentration. The values are 1.9774 and 1.9567. The explanation to the behavior constant or slow on hour 144 is due where there is already a decrease in the consumption of substrate, as can be seen in Figure 3A, so the behavior coincides that surely the disposition of the diluted substrate was less available for the microalgae, that is why its exponencial growth stopped going up and there was not even a marked exponential phase as observed in Figure 1B. In the same way, this refers to the effect of the photoperiod as it is an inducing factor of metabolic activity for cell growth. In the case of the 0.0005 g/L concentration, it can be seen that at hour 144 the substrate availability is minimal in Figure 3A, so this low value may be due to the lack of consumption of sodium acetate and that by so growth is not promoted.
  6. Figure 1B: I don't totally agree with your observation of the latency phase, the exponential phase and the stationary phase. The highest value was obtained at hour 144 or 216 (?). In the text, line 110, there is a problem with values, especially with the value 1.71/ Correction: if there was an error in the value 1.71, the values are «For the photoperiod of 16 hours of light at the same time, 168 is where the highest value of Log10 is found, being from; 2.37, 2.37 and 2.52 for the concentration of 0.0005, 0.001 and 0.0015 g/L respectively». The error was corrected. The highest value was at hour 144, which was specifically 2.62075 for the sodium acetate concentration of 0.0015 g/L. At hour 216, the value was 2.6349. It is generally mentioned that it is the highest value reached at hour 144 even though for the condition of 0.0005 g/L and 0.0010 g/L it is found at another time because it must be compared to the same hour where that point has been found. Furthermore, at hour 144 at the value of 2.62075 when exponential growth no longer occurs, or increasing, the behavior becomes slightly variable so it is taken as the end of the exponential growth phase.
  7. The interpretation of figure 1A-B needs to be reviewed/ Correction: the Figure was analyzed and add the necesary coments to be clear. We put cell growth.

Figure 1 C/D.

  1. Line 134 to 138: There is probably an inversion « With the Tukey-Kramer HDS analysis, it was possible to identify that the glycerol conditions at a concentration of 0.33 and 0.44 g/L at a SHORT photoperiod (8 hours of light) and the condition of 0.44 g/L at a PROLONGED photoperiod(16 hours of light), are not significantly different since there was no cell division. However, the condition that favors growth and division is that of 0.33 g/L of glycerol in prolonged photoperiods/ Correction: in line 134 to 138: The sentences was changed to « With the Tukey-Kramer HDS analysis, it was possible to identify that the glycerol conditions at a concentration of 0.33 and 0.44 g/L at a SHORT photoperiod (8 hours of light) and the condition of 0.44 g/L at a PROLONGED photoperiod(16 hours of light), are not significantly different since there was no cell division. However, the condition that favors growth and division is that of 0.33 g/L of glycerol in prolonged photoperiods».
  2. Line 138: «glycerol in prolonged photoperiods. As well as it can be seen»/ Correction: in line 138 was corrected the sentences to « glycerol in prolonged photoperiods. As well as it can be seen».

FIGURE 2

  1. - Line 151: To overcome this problem, the biovolum (µm3 /mL) should be obtained using a particle counter, especially for this microalgae OR you can do the same protocol (dry biomass) but with a higher volume of culture (ex 20 mL). I think the initial concentration is too high to see a real impact on the growth so on the biomass. Unless I am mistaken, you start with 5.07 cells/ mL and you have 2.108 cells/mL at the end of the experiment i.e. only 2 doublings of the population/ Correction: It was started with a concentration of 5.07 because in exploratory studies at a concentration of 1.07, 2.07 and 3.07 cells/mL, cell growth in the Neubauer chamber could not be quantified. However, with a concentration of 5.07 cells/mL it was possible to have constant readings consistent with the consumption of substrate and the synthesis of metabolites in Figures 3, 4 and 5. As for only the duplication in cell growth, this may be mostly promoted as in other works mentioned in the introduction and in the discussion of results but specifically, with the work of Kong et al., it is sought that at low concentration of substrate there is a greater production of pigments with antioxidant activity, therefore , growth was only duplicated due to substrate limitation but there was intracellular antioxidant activity.
  2. Line 155: It is imperative to make the culture as homogeneous as possible before sampling for analysis. Homogenisation can be done manually/ Correction: in line 155: the samples were homogenized in each taking to avoid agglomerations or clusters in the microalgae, however it has been observed that as the number of cells increases this is very common and can have an effect on the measurements, despite that it was that the samples were homogeneous both by manual and mechanical emotion.
  3. Line 165: «0.0015 g/L of substrate, respectively.»/ Correction: in line 165 was corrected the sentence to «0.0015 g/L of substrate, respectively.».
  4. Why you don't mention figure 2C and D (glycerol results)? / Correction: we talk about the Figure 2C and 2D.

FIGURE 3

  1. Line 182 to 187: here is my proposal: «As can be seen in Figure 3A and 3B there was a significant consumption in the different concentrations of substrate. In figure 3B a greater decrease can be seen. This behavior is in agreement to that observed in Figure 1B, where it was the experimental condition with a greater formation of cells. In figure 3A the decrease was less pronounced, as well as the minor increases in cells of Figure 1A.» / Correction: in line 182 to 187 the sentence was changed to «As can be seen in Figure 3A and 3B there was a significant consumption in the different concentrations of substrate. In figure 3B a greater decrease can be seen. This behavior is in agreement to that observed in Figure 1B, where it was the experimental condition with a greater formation of cells. In figure 3A the decrease was less pronounced, as well as the minor increases in cells of Figure 1A.»
  2. Line 193 and 199: deletes a point/ Correction: in line 193 to 199: the points were deleted.
  3. Line 196 to 199 : «in Figure 1C.» The rest of the sentence, please rewrite/ Correction: in line 196 to 199 the style to describe the Figure 3 was corrected like the 18 point above mencioned.

FIGURE 4/5

  1. Line 225: that This -> that this/ Correction: in line 225 in the words “that This” was changed to “that this”.
  2. Line 256: 0.001g/L, are you sure?/ Correction: in line 256, 0.001 g/L, are you sure? Yes, i do because we are working with a biologic model and is posible have variations but, if Pearson´s correlation is near or major of 0.85 is accepted in stadistic analysis, less value there is not correlation.

Point 10:

III. DISCUSSION

Line 270: In your results, you say that the exponential phase starts at 96 h for sodium and for glycerol.

- Line 284-285: Can you check the numbers 3.8 g/L? You don't give this numbers line 166 in your results.

- Line 285: for the glycerol which quantity of biomass for 0.44 g/L concentration?

- Line 293: , .

- Line 295: g / L g/L (no space)

- Line 296 : Can you check the numbers please?

Response 10:

Line 270: In your results, you say that the exponential phase starts at 96 h for sodium and for glycerol/ Correction: in this line we talk about the exponencial and stationary phase to glycerol «Talking about glycerol condition, it can be seen that only under the condition of 0.27 g/L at a photoperiod of 8 hours of light and 16 hours of darkness was there significant growth, beginning the exponential passage at hour 96 and reaching the stationary phase at hour 168. Regarding the photoperiod of 16 hours of light and 8 hours of darkness, there was growth both in the 0.27 and 0.33 g/L conditions,... »

Line 284-285: Can you check the numbers 3.8 g/L? You don't give this numbers line 166 in your results/ Correctión: we correct the data and it is specified in the results at what time the value of 3,825 g/L of biomass is obtained.

Line 285: for the glycerol which quantity of biomass for 0.44 g/L concentration?/ Correction: was add the value in the text « followed by the 0.44 g/L glycerol condition with 1.8 g/L biomass concentration at hour 168...».

  1. Line 293:, ->./Correction: in line 293 , was changed to .
  2. Line 295: g / L -> g/L (no space)/ Correction: in line 295 the spaces were erased in g/L. In all the document.
  3. Line 296 : Can you check the numbers please?/ Correction: in line 296 the numbers were checked.

Point 11:

  1. MATERIALS AND METHODS}

- Line 349: What is the volume of culture? in an Erlen of what capacity?

- Line 354: «stock solution of P-IV», at what concentration?

- Line 359: «red LED», to which PAR (intensity)?

I will discuss the use of red light in the introduction. This is an important point. The spectrum of the led in the

supplementary information would be a plus.

- Line 356: 10x106 cells/ml 107 cells/mL

- Line 363: delete space: 0.05%

- Line 369: 50 x 10 6 cells / ml 5.107 cells/mL

- Line 359 and 371 and 379: 60 ° C 60°C (same at several locations)

- Line 384: the calibration curve Different solutions the calibration curve different solutions

- Line 385 and 401: 1M 1 M

- Line 402: 20 Mm 20 mM

- Line 408: rea-gents reagents

- Line 417 to 419: « Regarding DPPH, it is a technique in which the synthetic radical is stabilized by lipophilic

molecules such as terpenoids or alkalis. However, it has been reported that C. vulgaris, through

- Line 411 to 422: Please rework this paragraph to make it easier to read

- Line 422 to 425: For this, the cell rupture was carried out with glass beads by carrying out previous washes

with deionized water under centrifuge at 4000 rpm during 5 minutes. This operation was run three times to

eliminate the salts dissolved in the medium.

- Line 422: Go to the line before «

- Line 422: Go to the line before « »

To be clearer:

Antioxidant activity

For this, the cell rupture was carried out with glass beads by carrying out previous washes with deionized

method

equivalents of Trolox g/L.

In the DPPH method

Response 11:

  1. Line 349: What is the volume of culture? in an Erlen of what capacity?/Correction: in line 349, what is the volumen of culture? 200 mL medium with the concentrations mentioned in final volume with macro and microelements. The solution is put in a 250 mL Erlenmeyer flak capacity.
  2. Line 354: «stock solution of P-IV», at what concentration?/ Correction: the concentration was specified «P-IV [Na2EDTA (2 mM), FeCl3.6H2O (0.36 mM), MnCl2.4H2O (0.21 mM), ZnCl2.6H2O (0.037 mM), CoCl2.6H2O (0.0084 mM) and Na2MoO4.2H2O (0.017 mM)].»
  3. Line 359: «red LED», to which PAR (intensity)? I will discuss the use of red light in the introduction. This is an important point. The spectrum of the led in the supplementary information would be a plus/ Correction: the light kind was mentioned and we put the light intensity worked. The intensity is 405 luxes.
  4. Line 356: 10x106 cells/ml 107 cells/mL/ Correction: in line 356 10x106 cells/ml was changed to 1.07 cells/mL.
  5. Line 363: delete space: 0.05%/ Correction: in line 363 the spaces were deleted in each %.
  6. Line 369: 50 x 106 cells / ml 5.107 cells/mL/ Correction: in line 369, 50x106 cells/ml was changed to 5.107 cells/mL.
  7. Line 359 and 371 and 379: 60 ° C -> 60°C (same at several locations)/ Correction: in line 359, 371 and 379, the spaces in each °C were erased.
  8. Line 384: the calibration curve Different solutions the calibration curve different solutions/ Correction: in line 384, the sentence was changed to «the calibration curve different solutions».
  9. Line 385 and 401: 1M -> 1 M/ Correction: in line 385 and 401, 1M was changed to 1 M. Were put spaces.
  10. Line 402: 20 Mm -> 20 mM/ Correction: in line 402, 20 Mm was changed to 20 mM.
  11. Line 408: rea-gents -> reagents/ Correction: in line 408, rea-gents was changed to reagents.
  12. Line 417 to 419: « Regarding DPPH, it is a technique in which the synthetic radical is stabilized by lipophilic molecules such as terpenoids or alkalis. However, it has been reported that C. vulgaris, through...»/ Correction: in line 417 and 419, te sentence was changed to «Regarding DPPH•, it is a technique in which the synthetic radical is stabilized by lipophilic molecules such as terpenoids or alkalis. However, it has been reported that C. vulgaris, through... »
  13. Line 422 to 425: te sentence was changed to «For this, the cell rupture was carried out with glass beads by carrying out previous washes with deionized water under centrifuge at 4000 rpm during 5 minutes. This operation was run three times to eliminate the salts dissolved in the medium. ».
  14. Line 422: the sentences was changed to the indications by revisors.

Point 12:  

  1. CONCLUSIONS

Response 12:

  1. Line 460 to 462: «With the work carried out, it can be concluded that C. vulgaris is a species of microalgae that can grow favorably at the flask level under photoheterotrophic growth with sodium acetate or glycerol as substrate. However, its growth is favored when»/ Correction: in line 460 to 462, the sentences was changed to «With the work carried out, it can be concluded that C. vulgaris is a species of microalgae that can grow favorably at the flask level under photoheterotrophic growth with sodium acetate or glycerol as substrate. However, its growth is favored when»

Reviewer 2 Report

The current study describes the evolution of antioxidant production in Chorella vulgaris in vitro culture (cultivated in flask).

The topic is sound and of broad interest.

However, some technical questions remain after reading this paper, submitted to Molecules.

1) The mechanisms of ABTS and DPPH are very similar. Why did you choose these two tests?

Please add a clear justification and add discussion about the mechanism, and future perspectives.

2) One of the limitations of the current work for publication in Molecules, in my opinion, is the lack of phytochemical analysis.

The paper is appealing, but in the absence of any phytochemical analysis, I'm not sure it fits within the scope of molecules.

Author Response

Point 1: The mechanisms of ABTS and DPPH are very similar. Why did you choose these two tests?  Please add a clear justification and add discussion about the mechanism, and future perspectives.

Response 1:

The identification tests for secondary metabolites with antioxidant activity can be several, among which, and the most common, are the radical test of ABTS, DPPH, FRAP and ORACA. However, for this work it was decided to use the ABTS and DPPH tests because they are simple tests, but highly reliable and although the mechanisms are similar, it is important to mention that the ABTS test is for lipophilic and hydrophilic metabolites. , that is, polar and non-polar, this is reaffirmed since there is evidence that it also avoids unwanted secondary reactions and does not require high temperatures to generate free radicals, it also avoids interference by endogenous peroxidase activity, so it can be determined in plant samples and other extracts [1]. In the case of the DPPH method, we can also refer to this foundation with the work of Cheng et al., which reaffirms and supports the decision to use these two methods [4].

Point 2: One of the limitations of the current work for publication in Molecules, in my opinion, is the lack of phytochemical analysis. The paper is appealing, but in the absence of any phytochemical analysis, I'm not sure it fits within the scope of molecules.

Response 2:

Phytochemical analysis is a perspective that is taken into account, however, for now it has focused on the analysis of growth and whether or not there is antioxidant activity. However, there are reports that the antioxidant activity detected in plant materials is mainly due to the pigments contained and as discussed in the document, C. vulgaris contains pigments. Within these works the following can be mentioned; "Antioxidant capacity of five mango cultivars (Mangifera indica L.) and evaluation of their behavior in a food matrix", in which the concentration of β -carotene and the antioxidant activity with the ABTS•+ [3] radical are evaluated, also There is the work entitled "Obtaining chlorophyll from aloe (Aloe barbadensis) husks by means of solvents" in which the content of chlorophylls and antioxidant activity were determined by the ABTS•+ and DPPH• method, observing a relationship with form a its presence and it is also evaluated through these two methods for the same reason indicated in answer 1[6]. On the other hand, we can also mention the work entitled "Effect of the use of Ascophyllum nodosum extracts on the growth, production and antioxidant activity of carotenoids from Haematococcus pluvialis" in which the presence of β-carotene in the microalgae is determined and they measure antioxidant activity using the DPPH• method, so with this it can be seen that this method is lipophilic since the nature of carotenes is non-polar[5]. Finally we will mention the work entitled "Two-stage cultivation of Chlorella vulgaris using light and salt stress conditions for simultaneous production of lipid, carotenoids, and antioxidants" in which a phytochemical analysis is made mentioning the presence of pigments such as carotenes and the antioxidant activity measured. by the ABTS•+ and DPPH• method and a relationship between the measured responses can be observed [2]. With all of the above, it can be assumed that these pigments may be present in C. vulgaris analyzed in this work and therefore continue with future research to analyze the effect of stress conditions with respect to the percentages of production of lipophilic and hydrophilic metabolites.

  1. Alcolea JF, Cano A, Acosta M, Arnao MB. Hydrophilic and lipophilic antioxidant activities of grapes. Food/Nahrung 2002, 46(5), 353-356.
  2. Ali HEA, El-fayoumy EA, Rasmy WE, Soliman RM, Abdullah MA. Two-stage cultivation of Chlorella vulgaris using light and salt stress conditions for simultaneous production of lipid, carotenoids, and antioxidants. Journal of Applied Phycology 2021, 33(1), 227-239.
  3. Arrazola G, Rojano BA, Díaz A. Capacidad antioxidante de cinco cultivares de mango (Mangifera indica L.) y evaluación de su comportamiento en una matriz alimentaria. Revista Colombiana de Ciencias Hortícolas 2013, 7(2), 161-172.
  4. Cheng Z, Moore J, Yu L. High-throughput relative DPPH radical scavenging capacity assay. Journal of agricultural and food chemistry 2006, 54(20), 7429-7436.
  5. Guajardo-Barbosa C, Galán-Wong LJ, Beltrán-Rocha JC, Quintero-Zapata I, Gandarilla-Pacheco FL, Pereyra-Alferéz B, Luna-Olvera HA. Efecto del uso de extractos de Ascophyllum nodosum en el crecimiento, producción y actividad antioxidante de carotenoides de Haematococcus pluvialis. Revista de biología marina y oceanografía 2020, 55 (1), 79-84.
  6. Pacheco YEG, Stand LIM, Pinto NM, Palacio J, Angarita A, Vargas-Barrios D. Obtención de clorofila a partir de cáscaras de sábila (Aloe barbadensis) por medio de solventes. INGE CUC, 17(2).

Round 2

Reviewer 1 Report

Thank you for taking my remarks into account.

Best regards

Reviewer 2 Report

Thank you very much for providing such a detailed response to my queries. I really appreciate it, as well as the tone of this discussion. I recommend the paper for publication.